# Parents' online coaching in the early intervention home program for toddlers with autism spectrum disorder during the COVID-19 pandemic: Manual development and feasibility study

JieYi Png[1], Farahiyah Wan Yunus[1]*, Masne Kadar[1], Yang Wai Wai[2], Yazmin Ahmad Rusli[3], Jamilah Hanum Abdul Khaiyom[4]

1 Centre for Rehabilitation and Special Needs Studies, Occupational Therapy Programme, Universiti Kebangsaan Malaysia, Jalan Raja Muda Abdul Aziz, Kuala Lumpur, Malaysia, 2 Faculty of Medicine, Department of Pediatric, Universiti Kebangsaan Malaysia, Hospital Tunku Ampuan Besar Tuanku Aishah Rohani, Hospital Pakar Kanak-Kanak UKM, Jalan Yaacob Latif, Bandar Tun Razak, Cheras, Kuala Lumpur, Malaysia, 3 Centre for Rehabilitation and Special Needs Studies, Speech Sciences Programme, Universiti Kebangsaan Malaysia, Jalan Raja Muda Abdul Aziz, Kuala Lumpur, Malaysia, 4 Department of Psychology, International Islamic University Malaysia, Kuala Lumpur, Malaysia

* farahiyahwanyunus@ukm.edu.my

**Data Availability Statement:** The data supporting the findings presented in the study are included within the manuscript. However, access to the raw

## Abstract

Early intervention improves the developmental progress among toddlers with ASD. Family involvement enhances the intervention outcome. This study aimed to develop and test the feasibility of an early intervention home program manual for toddlers with ASD. Method: This study involved three phases: (I) formulation of manual concept and content design (II) manual development through focus group discussion (n = 10) and content validation by experts (n = 9); (III) cognitive interview (n = 6) and feasibility study (n = 8). Result: Content Validity Index (I-CVI) for the developed manual ranged from .78–1.0, S-CVI/Ave .96, and S-CVI/UA .79. Cognitive interview among six parents reported that the manual was easy to understand. The feasibility study reported all eight parents benefitted from coaching sessions. Approximately 87.5% of the respondents found the program benefited their children and could easily implement the activities in their daily routines. Approximately 75% of respondents reported having greater confidence in managing their child's behaviors at home. Parent coaching using the developed home program is feasible and accepted by parents of a toddler with ASD. Further study should be developed to explore the effectiveness of parent coaching using the manual.

## Introduction

The Autism Spectrum Disorder (ASD) diagnosis rate has been increasing over the years. For 2014, it was reported as one in 59 children and recently for 2020, it has been reported as one in 36 children across 11 sites in the United States, among 8 years old children, while the gender

datasets is restricted due to ethical considerations. For inquiries or requests regarding access to the raw data for further analysis or future studies, please contact the ethical committee Ms Norsuriati Mustapha via email at sepukm@ukm.edu.my or via telephone at +603-9145 5046 / 9145 5048 for researchers who meet the criteria for access to confidential data. The ethical committee's website can be accessed at https://www.ukm.my/jepukm.

**Funding:** The study was funded by Geran Galakan Penyelidik Muda (GGPM-2020-023) Universiti Kebangsaan Malaysia Grant and the Penyelidikan Strategik Translational (TR@M) Translational Grant (TRANSLATIONAL-2019-001/2). The funders had no role in study design, data collection and analysis, decision to publish, or preparation of the manuscript.

**Competing interests:** NO authors have competing interests.

ratio remains at four males to one female [1,2]. According to DSM 5-TR, ASD is a neurological condition that presents challenges in social interaction, reciprocal communication, repetitive behaviors and limited interest [3]. Commonly, parents reported their main concerns about their children's delay in speech-language communication and social behavior challenges [3].

With the increase in early diagnosis of ASD, its signs and symptoms can be detected early in the first two years of life [3,4]. Hence, efforts in developing various early intervention programs are needed to support the needs of this population [5,6]. The aim of the early ASD diagnosis is to ensure toddlers with ASD can have access to early intervention as soon as needed [7]. Most parents raised concerns about their children not receiving diagnosis until the age of 36 months, which could be considered as late [1,3]. This is a concern since early intervention programs is only provided for children with special needs from infants to 36 months old [8]. Many early intervention programs have shown effectiveness in improving core symptoms of ASD, especially to those who started the intervention before 3 years of age [7,9,10]. Hence, it is highly recommended that early intervention to start as early as possible in order to gain its full benefits. Meanwhile, during the brain development of a toddler, neural connectivity and networking happen rapidly in the social interaction process [11]. Hence, reciprocal interaction and communication stimulation in early intervention are among the key aspects to focus on.

Most of the intensive early intervention programs carried out for 20–40 hours per week showed positive outcomes on the challenges in social, communication and adaptation skills in children with ASD [12,13]. Those interventions are mainly delivered directly to the children by trained professionals, without parents' direct involvement. Although the positive outcome correlated with the amount of hours invested in the intervention, the cost and manpower remain a challenge, especially to parents with low incomes [14,15]. In recent years, parent or family involvement in early intervention has been highlighted; a comprehensive intervention is having parents actively involved in the intervention and learning through daily routine activities [9,16,17]. Parent-child interaction is essential for early intervention by coaching parents to promote social interaction and communication with toddlers allows daily learning opportunities in their natural environment, as the toddler mostly lives at home. Early intervention promotes joint attention through play and interaction with parents, in which parents can generalize the learning concept into spontaneous parenting in the daily routine, instead of structured learning sessions [18].

Although 75% of parents recognize their child's developmental milestone delay before 3 years of age, nevertheless, the average time gap between parents' concern over their child's developmental challenges and getting a diagnosis is reported as 21.5months [3]. Spending more than one year in getting a confirmed diagnosis is an unfortunate delay that forfeits the opportunity for early intervention by the time diagnosis is made. Parent-mediated intervention for high-risk toddlers showed potential benefits in parent-child interaction, as well as in the aspect of motor and social development skills [16]. The cost of living for a family with individuals with special needs is higher compared to the regular population [19]. The stress levels of parents with special needs children are reported as higher than those with typically-developing children [20]. However, parents' stress levels were shown to be significantly lower when they received coaching session by relevant professionals to manage their children with special needs [21].

At the same time, when parents are able to perform the intervention at home themselves with good coaching sessions, this will make them less dependent on professional time commitment, such as taking work leave and other costs that may be involved in order to attend the clinic sessions regularly [18]. Therefore, home intervention reduces the cost and time burden on parents. Furthermore, a previous study on an early intervention program showed positive outcomes among 3–9 year-old children with ASD, and it also identified challenges including

behavior, environment, culture and belief in its implementation [22]. Moreover, most of the early interventions programs were based in a center or hospital settings, using specialized equipment and tools, which made it less practical for parents to carry out or repeat the intervention at home. Thus, there is a need to develop a parents' early intervention home coaching program that is suitable for the home environment. Hence, the aim of this current study is to develop, and determine the feasibility of a parents' coaching manual in the early intervention home program for toddlers with ASD.

## Methods

### Research ethics

Research ethic approval was obtained from the Medical Research and Innovation Secretariat, Universiti Kebangsaan Malaysia (project number: JEP-2020-757).

### Process

The early intervention manual development adopted the guideline from the study of Galinsky and the group [23]. The process includes three phases: (I) Formulation of the manual concept and content design, (II) Revision and validation of content by an expert panel, (III) Implementation through cognitive interview pretesting and feasibility study.

The process of data collection was impacted by the COVID-19 pandemic (March 2020-June 2021); therefore, the face-to-face data collection process had to adhere to strict standard operating procedures against COVID-19. Hence, most of the data collection in this study was done through online platforms, [24] which could be considered as the most suitable for the current situation.

**Phase I: Formulation of manual concept and content design.** After conducting a literature review in the field of early intervention for ASD, to our knowledge, it seems studies on parents' active involvement in early intervention programs for toddlers with ASD are still scarce [7,9,16,18,25,26]. Hence, culturally-adapted early intervention home program content was developed, based on the developmental and relationship needs of a toddler with ASD. All the domains and activities covered in the manual are based on the developmental milestones and adaptive behaviors from the Centers for Disease Control and Prevention (CDC) and the American Academy of Pediatrics (AAP) [27].

**Phase II: Content development and validation by an expert panel.** Between January—May 2021, ten panel experts were recruited, which is sufficient for the process, based on the recommendation of 6–10 members [28,29]. Purposive sampling is needed when recruiting the experts as their expertise related to the study is essential [30]. The inclusion criteria to be an expert panel member is having at least 5 years' of clinical working experience with children with ASD [29]. The consent form was provided and acknowledged through email upon the expert's agreement to participate. A soft copy of the manual and focus group discussion (FGD) agenda was emailed to each of the experts two weeks before the FGD session, with the intention that the experts review the developed program manual before the FGD session. The FGD session was done through an online platform on 3[rd] June 2021, for 2 hours 30 minutes with the author as a moderator, and the whole process was video recorded with permission.

After modification of the program manual based on the FGD results, the program manual was then progressed to content validation; a Content Validity Index (CVI) form was constructed and distributed to the experts. The CVI form consists of four validity criteria: (1) relevance, (2) simplicity, (3) clarity and (4) ambiguity [31,32]. Each of the criteria was scored on a Likert scale of 1 to 4, where 1 = not relevant, 2 = item needs amendment, 3 = relevant with

minor amendment, 4 = highly relevant. Open-ended questions were also included for additional comments from the experts on any items to be improved in each of the domains.

The CVI form, together with all the necessary information was then sent out. These included a summary of the research, consent form, demographic form and a copy of the program manual, with clear written instructions. A two-week period for completion of the responses was given with regular follow-ups to remind the experts of the deadline. As recommended by Polit et al. (2007) [28], the collected data were analyzed for two types of CVI: CVI for items (I-CVI) and CVI for scale (S-CVI). Two methods were used to calculate S-CVI. First, the S-CVI/Ave was derived by computing the average score of all the I-CVI items. Next, the S-CVI/UA was computed by calculating the proportion of scores that ranked 3 and 4 against the total of all 42 items. The authors reviewed all the feedback from the experts, including semantics and appropriate choice of words in contextual usage. Based on the feedback, several amendments were made to the program manual.

**Phase III: Implementation through cognitive interviewing (pretesting) and feasibility study.** After incorporation of feedback from the experts, the program manual was used in the cognitive interviewing process with a small selected group of parents who have toddlers with ASD to ensure that the target population was able to comprehend and use the manual effectively [33]. A sample of between five and fifteen respondents is recommended for that purpose. Accordingly, six parents from various ethnicities were recruited from one of the major teaching hospitals in Malaysia. The parents who have toddlers aged 32–35 months, diagnosed with ASD, were recruited from June to July 2021 via the Child Developmental Clinic (CDC) of the chosen hospital. Respondents were given the online consent and demographic form before the online cognitive interviewing session. After the collection of the signed online consent form, the program manual was given to the respondents for them to review four weeks prior to their interview sessions. Cognitive interviewing sessions were conducted from 28[th] to 31[st] July 2021, through the Zoom platform [24]. During the interview, respondents were asked about their understanding of the content; specifically, whether they understood what to do to conduct the activities at home, whether there were words that they did not understand or words they were confused about. Respondents were then requested to rephrase those items or words which were not clear. All responses were resolved with respondents within the average interview time of 45–60 minutes. The final version of the program manual was produced after consolidating feedback from parents and was then used in the next phase of the study; i.e., the feasibility study.

A feasibility study is essential to determine the suitability of the developed manual on a small scale of population [34]. For this purpose, parents who have toddlers aged 18–35 months with ASD were recruited on 1[st]– 30[th] August 2021, through purposive sampling among the name list from the chosen teaching hospital. Sample sizes for qualitative research in feasibility studies are typically small, usually between 5 and 20 individuals [35]. This is reasonable, as simulations indicate that 10 users can identify at least 80% of issues with the technology during usability testing, while 20 users can identify 95% of the problems [36]. Eligible parents were provided with general information about the study and the consent form online. Upon receipt of the signed consent and demographic information online form, the assessment date and details were given. The assessment was conducted face-to-face in a clinic setting in the chosen hospital. The assessment battery, which includes Vineland-3, Toddler Sensory Profile 2-Malay version (TSP 2-M) and Quality of Life in Autism Questionnaire (QoLA), was conducted within 75–90 minutes in one session with a break in between. Due to COVID-19 SOP, only one completely vaccinated parent and his/her toddler with ASD were allowed to attend the session; the same parent was assigned to be the main caregiver to attend the online coaching sessions. A printed version of the program manual was given out, followed by four weekly online

coaching sessions of 60 minutes each session between 31st August and 30th September 2021. After the fourth coaching session, a semi-structured interview was conducted with a face validity form. In order to analyze the feasibility of the program manual, the data from the semi-structured interview were then analyzed thematically through a deductive approach [37,38]. Utilize the qualitative approach is common to assess the acceptability and feasibility of the developed program manual as an intervention [35,38].

## Data analysis/result

### Result on phase I of the study

Table 1 shows the content of the parents' early intervention home coaching program. The content includes five chapters, and the activities are tailored to the expected developmental domains and milestones, as well as adapted to local culture and needs. Early intervention mainly focuses on supporting the ASD core deficits, which are social interaction and communication.

Ten professionals were invited to form the expert panel during the FGD session; they included three occupational therapists, one speech and language therapist, two clinical psychologists and four special needs educators, two of whom are parents of children with ASD themselves. Table 2 shows the demographic information of the expert panel, with a minimum of nine years and a maximum of thirty-two years of working experience.

Qualitative data were analyzed using the thematic analysis method, and video and audio recording were transcribed in Word. To ensure the accuracy of the findings, the results and amendments were shared to all expert panel members for review during a member checking process.

**Table 1. Content of the early intervention home program.**

| |
|---|
| Chapter 1 Introduction to Home-based Early Intervention Program |
| 1.1 What is Early Intervention Program |
| 1.2 What is Autism |
| 1.3 Objectives of program |
| Chapter 2 Developmental Milestone from 18–36 months |
| 2.1 Gross motor and fine motor skill |
| 2.2 Sensory adaptive behavior |
| 2.3 Communication and speech |
| 2.4 Social emotion |
| 2.5 Activities of daily living skill |
| 2.6 Play skill |
| Chapter 3 Early Intervention Home Program Concept |
| 3.1 Parents involvement |
| 3.2 Natural learning environment |
| 3.3 Routine-based activities |
| 3.4 Development and relationship-based learning |
| Chapter 4 Activities Manual |
| 4.1 Activities guideline |
| i. Gross and fine motor skills |
| ii. Sensory/senses |
| iii. Communication and speech |
| iv. Social emotion |
| v. Daily activity skill |
| Chapter 5 Strategy in Dealing /Handling Children with Autism |
| 5.1 Emotional regulation |
| 5.2 Simple and direct response |
| 5.3 Minimized use of don't /no |
| 5.4 Reward system |

**Table 2. Demographics of the expert panel in FGD.**

| Reviewers | | N = 10 (%) | Mean (SD) |
|---|---|---|---|
| Age group (years) | | | 43.4 (8.4) |
| Gender | Male | 1 (10.0) | |
| | Female | 9 (90.0) | |
| Academic qualification | Diploma | 2(10.0) | |
| | Bachelor's | 3 (30.0) | |
| | Master's | 4 (40.0) | |
| | PhD | 1 (10.0) | |
| Working experience (years) | 5–14 | 5 (50.0) | 17.1 (8.6) |
| | 15–24 | 3 (30.0) | |
| | >25 | 2 (20.0) | |

The discussion was concluded in three main themes–Theme I: Parents' role in early intervention; Theme II: knowledge and resources to support early intervention; Theme III: Suitability of activities in the early intervention home program manual.

**Theme I: Parents' role in early intervention.** Nowadays, most families have both parents working, and the arrangement of giving childcare duties to the nanny, nursery or relatives would help to ease the parents' daily burden. When a child is diagnosed with ASD, parents play an important role in intervening and support the child's developmental needs. Helping parents to identify their priority and their capacity for intervention participation would support the early intervention outcome. These opinions were among the main concerns expressed by professionals involved in the FGD session; for example, according to one expert: *"parents can be confused. . . they might need some guidance to clarify their concerns and confusion. . . I suggest you use the Canadian Occupational Performance Measure (COPM) to guide them, give them choices and invite their participation based on their availability in the daily routine."*

Most of the experts believed that the barrier to active parent participation is most likely the logistical arrangement as well as time constraints, as one expert suggested: *"both are working parents. They might not be able to get involved too much. . . might not have much time and opportunity to play with and teach the child."*

**Theme II: Knowledge and resources to support early intervention.** All experts agreed with the inclusion of basic information about ASD, in order to help parents better understand their children's conditions; for instance, the ASD characteristic, main challenges and intervention approaches and concepts that support ASD. One expert said: *"Knowledge about ASD and neurodiversity helps parents to understand the context and provides them with a bigger picture for future planning."*

Each domain in the developmental milestones provides guidance and expectations for parents, which we believe would help parents in intervention plans. *"The autism population mainly struggles with expressive skills and challenging behaviors, so the checklist of developmental milestones could help parents to work on it. . . prioritize the main concerns appropriately,"* was how one of the experts expressed it. Additionally, empowering parents to seek further professional support, when necessary, should be included. As one expert explained: *"The manual is not a total replacement of the interventions. Parents should always look for trained professionals for continuous support."*

**Theme III: Suitability of activities in the early intervention home program manual.** The expert panel had a few suggestions for the content, which include the suitability of the selected activities and the strategies involved. It is important to ensure consistency of the

**Table 3. Demographics of the expert reviewers for CVI.**

| Reviewers | | N = 9 (%) | Mean (SD) |
|---|---|---|---|
| Age group (years) | | | 42.7 (7.0) |
| Gender | Male | 1 (25.0) | |
| | Female | 8 (75.0) | |
| Academic qualification | Diploma | 2(12.5) | |
| | Bachelor's | 2 (50.0) | |
| | Master's | 4 (37.5) | |
| | PhD | 1 (10.0) | |
| Working experience (years) | 5–14 | 5 (62.5) | 15.4 (8.8) |
| | 15–24 | 3 (37.5) | |
| | >25 | 1 (20.0) | |

illustration-related titles and emphasize the ASD characteristic when stating the examples, as one expert mentioned: *"The activities can be used for any other child, giving examples of ASD-related behaviors, such as rigidity in a certain way during activities and recommend methods to support it."* The use of flowcharts for activities, as well as details of each activity based on the developmental age group were also recommended by the panel. Activity suggestions should consider families in the lower financial economic status, who might have very limited space at home, limited receptive language ability, or limited resources. Hence, the activities' adaptation and presentation can be varied by using common household tools/gadgets that can be easily acquired. One expert suggested: *"Having a child use a laundry basket is fun. Suggest the best time for the parents to use it and build it into the routine. This will also help to improve the parent-child bonding."*

## Result on phase II of the study

The same ten professionals were invited to review the home program manual that had been improved, based on the previous FGD session, so that they could give ratings on the content of the manual; nine experts completed the CVI form. Table 3 shows the demographic information of the expert panel involved; each had between nine and thirty-two years of working experience, with an average of 17.1 years of working experience. Face validity results showed 100% appropriateness of the font size, font types and content arrangement. All expert panel members indicated high agreement on the content of the manual. In respect of I-CVIs, out of the total of 43 items of the program manual, 34 items scored 1.0, while the remaining 9 items (item 3 and 4 from Chapter 2; item 2 and 4 from Chapter 3; item 3, 4 and 12 from Chapter 4; item 1 and 5 from Chapter 5) were scored between .78 and .89. All I-CVI values were above the acceptable range as proposed by Lynn (1986), which should be a minimum of .78, if scored by at least nine experts. In terms of S-CVI, the usual lowest acceptable value is .80 [31] The program manual has S-CVI/Ave = .99 and S-CVI/UA = .89 for all four criteria (relevance, simplicity, clarity and ambiguity). The higher score of the CVI (1.0) indicated the panel's agreement on the contents. The result of I-CVI, S-CVI/AV and S-CVI/UA values met satisfactory level, (>.78)Thus, this indicates the manual content adequately represents the intended construct it is meant to assess [30–33].

## Result on phase III of the study

Table 4 shows the demographics information of six parents involved in the study for the cognitive interviewing process among mothers of toddlers with ASD. All cognitive interviewing

Table 4. Demographics of the parents and special needs children in cognitive interview.

| Characteristics | Result | N = 6 (%) | Mean (SD) |
|---|---|---|---|
| Parent's age (years) | 25–29 | 1 (16.7) | 32.7 (4.9) |
| | 30–34 | 3 (50.0) | |
| | 35–39 | 2 (33.3) | |
| Parent's gender | Male | 2 (33.3) | |
| | Female | 4 (66.7) | |
| Children's age (months) | 32 | 1 (16.7) | 33.3 (1.4) |
| | 33 | 2 (33.3) | |
| | 34 | 3 (50.0) | |
| Children's gender | Male | 5 (83.3) | |
| | Female | 1 (16.7) | |
| Parent's education level | Diploma | 3 (50.0) | |
| | Degree | 3 (50.0) | |
| Parent's race | Malay | 4 (66.7) | |
| | Chinese | 1 (16.7) | |
| | Indian | 1 (16.7) | |

sessions were carried out virtually through video conference via the Zoom platform. Overall, the interview sessions went smoothly, despite some challenges with technical network connections from interviewer and interviewee sites. The cognitive interviewing indicated that all of them (100%) agreed on the size and type of the font, the content layout of the program manual, as well as the clarity of the items. Data indicated that all words and phrases in the manual were clear and easily understood by parents and the illustrations were helpful when planning for the activities.

**Qualitative descriptive analysis of the background and feasibility study.** Fifteen candidates from the chosen teaching hospital were found to fulfill the inclusion criteria and were invited to be involved in the study. Eight parents provided consent and completed the study under strict COVID-19 SOP. All eight parents participated in four-week online coaching sessions with 100% attendance. Table 5 shows that their demographic information consists of five mothers (62.5%) and three fathers (37.5%) who live in Klang valley in Malaysia.

Thematic analysis of the semi-structured interviews found three main themes were explored with parents: (I) Knowledge and ideas for parents involved in the early intervention home program; (II) Application of techniques or strategies in the daily activities routine; (III) feasibility of conducting suggested activities at home. The processed results were reviewed by the parents during member checking process. Face validity showed that the font and sizes used were suitable, content arrangement was clear, the words and phrases in the manual were simple and easy, and the illustrations were helpful. All parents (100%) reported that they benefited from the online coaching sessions based on the manual and that the manual is easy to understand.

For knowledge and ideas for parents' involvement in the early intervention home program, the lack of understanding about ASD and the child's current developmental needs can lead to parent's frustration and unrealistic expectation. Sometimes the child with ASD tended to walk away or scream, which disrupted others. The parents' role as co-regulator is essential. One of the parents gave feedback on her helplessness when she was trying very hard to engage the son but to no avail. During the online coaching sessions, the mother was taught and supported in how to slow down on the steps involved in the activities and to wait for her son to respond. Subsequently, the son started to respond to her unexpectedly. All parents mentioned they

**Table 5. Demographics of the parents and special needs children in the feasibility study.**

| Characteristics | Result | N = 8 (%) | Mean (SD) |
|---|---|---|---|
| Parent's age (years) | 25–29 | 1 (12.5) | 36.1 (4.8) |
| | 30–34 | 2 (25.0) | |
| | 35–39 | 3 (37.5) | |
| | 40–44 | 2 (25.0) | |
| Parent's gender | Male | 3 (37.5) | |
| | Female | 5 (62.5) | |
| Children's age (months) | 32 | 3 (37.5) | 33.4 (1.3) |
| | 33 | 1 (12.5) | |
| | 34 | 2 (25.0) | |
| | 35 | 2 (25.0) | |
| Children's gender | Male | 5 (62.5) | |
| | Female | 3 (37.5) | |
| Parent's education level | High School | 1 (12.5) | |
| | Diploma | 4 (50.0) | |
| | Degree | 3 (37.5) | |
| Parent's race | Malay | 7 (87.5) | |
| | Chinese | 1 (12.5) | |

obtained better activity ideas to conduct the home program. Parents acknowledged the clarity and goals of the activities and learned to use daily objects at home to support the child. Comments from parents included: *"Now I know a laundry basket can be used to engage my child as therapy too. I have to do laundry anyway, so I can ask her to help me,"*... *"I discovered how to use the flour to play with my kid. It's so much fun when I don't control them... I follow the way they play."*

When applying strategies in parenting and interaction, parents shared their gains and doubts during the online coaching sessions. Parents were empowered to support their children's daily communication through the activities. As one parent mentioned: *"I learned to stop myself from asking him to answer repeatedly, to pause. I need to be more patient... he is learning at his own pace."* When parents change, the child might change indirectly, as adults learn, and put into action as a model for the child to learn. One mother summed it up: *"I try to change my way of communicating with him. I used to say, 'No, don't run, don't climb,'... I have to tell him what he is supposed to do; it's not easy for me to remember to say that."* It is particularly difficult to follow the child's lead when adults have expectations in mind. The importance of connection prior to any intervention helps smooth the intervention engagement and goal achievement. As one parent stated: *"I was so anxious. I want him to listen to me; I don't want him to cry and scream. It's so embarrassing but, now I have learned to read the signs from my child, I try not to force him and go along with him, depending on his mood... then I will add in demands only when he is ready."* From the questionnaire, six of the parents (75%) say they have the confidence to manage their child's behaviors at home with the online coaching sessions.

With regard to the feasibility of conducting suggested activities at home, the concept of using household items and any adapted tools from home has inspired the parents' creativity to play with their child. Many parents have found that reading books to toddlers with ASD is challenging, as the toddlers cannot sit still. However, when the strategies were recommended, parents noticed the changes in their children, as indicated by one mother: *"I can see improvement in my child; he settles down faster ... I can see him looking at the pictures."* Parents received various information and ideas; however, without proper coaching, such situations

may sometimes hinder parents from initiating the home program effectively. With guidance, the parent utilizes available tools at home to execute the intervention step by step. One parent made this clear: *"I have no idea where to start and how to start. I know I should do something but I feel really tired. . . having this online coaching session has helped me to start with something. I can do some activities with my boy at home now."* The results from semi-structured interviews indicated that parent coaching using the developed home program is feasible and accepted by the parents of toddlers with ASD. One of the suggestions given was to prepare an activity video as a reference for busy parents.

## Discussion and study limitation

Feasibility studies help identify and rectify practical issues, refine research protocols, and ensure methodological rigor. This feasibility study aims to estimate key parameters for designing a primary randomized controlled trial and explicitly describe the intervention for practical application. Lack of testing in an intervention can result in it being impractical, inapplicable, and not addressing the needs of practitioners and clients. This may also lead to the inappropriate use of instruments, failing to measure outcome changes accurately.

This study identifies the feasibility of the developed parent online home coaching program as an early intervention for toddlers with ASD and explores parents' acceptance and program implementation at home. The high level of acceptance from parents' involvement in intervention is consistent with the studies [39,40]. Parent and family involvement in early intervention, natural setting at home and routine-based intervention yield various positive outcomes for children and families [41]. Hence, based on the approaches adopted by the manual, it is feasible to be used by parents of toddlers with ASD.

The attendance rate is high (100%), but this might be due to the selective nature of the research samples and the level of motivation of the parents involved in the study. Retention was good, no dropout. The parents are ready and keen to learn and support their children due to the restricted circumstances created by the COVID-19 preventive health measures, there was a shortage of available intervention for ASD due to the closure of centers and suspended hospital outpatient appointments [42]. Most parents have to stay at home or work from home, and having a toddler with ASD with limited resources can be stressful. Thus, having online coaching support might fulfill some critical needs of the parents and their children with ASD during the pandemic. Similar findings from systematic reviews suggested that coaching parents in intervention via online is feasible [43].

The application of an early intervention home program and the received support through online coaching sessions guided parents to utilize materials, tools and toys available at home to run the session [42], which was convenient for the parents, without additional logistic arrangements and beneficial to the family, along with the benefit of being cost-effective [26,41]. Such online coaching sessions on the home program may overcome the challenges of having limited trained professionals to perform face-to-face intervention, as well as challenges on the high cost of intervention when accessing an early intervention program, especially in developing countries [14,15,48,49]. Parental training and parent-mediated intervention for children with ASD involving trained professionals suggested improvement in social interaction and quality of life of the families [44–48].

In line with the findings of this study, a recent scoping review explored the feasibility of training parents of children with ASD in limited settings, as defined by the United Nations, to administer interventions to their children within their natural home environment. Despite limited research availability, the results echoed those of this study, suggesting that home-based intervention could be a practical approach to supporting families in limited resources settings

[48]. A noteworthy discovery was the significant correlation between culturally tailored interventions and parents' readiness and capability to effectively implement them [42]. Home-based interventions are feasible and accepted by parents of a toddler with ASD in the study.

However, some limitations were faced during the online coaching session, especially in managing the technical issues which arose from poor internet connectivity, both on the researchers' as well as participants' end [44]. However, researchers took the necessary steps to ensure the data quality, such as making a voice/video call to explain and clarify written instructions to parents prior to the administration of the activities, and by being available online for parents when they had inquiries during the process, thereby ensuring that parents had a good understanding of the home program manual. Data collection through online platforms holds great promise for advancing ASD research by enhancing accessibility and flexibility, However, ethical, valid, and reliable data collection practices must be carefully considered.

As a feasibility study, the current study faced several key limitations. Firstly, the participant sample size was small and skewed towards urban and educated parents. The study was conducted in Klang Valley, Malaysia's main economic hub, chosen for its logistical convenience. However, its socioeconomic and cultural diversity may not fully represent the entire country, as rural families likely have different experiences, beliefs, and needs. Data saturation should be adopted to ensure the robustness of the study [49]. Having more parents involved in the initial concept development would have been beneficial. Additionally, the quantitative parent and child outcomes have not yet been compared to a control condition.

The findings from this study guide further adjustments of the manual and expansion of the study. For example, more emphasis has been placed on active family routine tasks, coaching, and discussion with parents, rather than on didactic teaching in the session. The activity manual has been further refined, additional focus on guiding child participation and interactive communication with family members [50].

By leveraging the strengths of online platforms while addressing potential limitations, harnessing the full potential of technology to improve our understanding of ASD and inform interventions and support services for individuals and families affected by this complex condition [44–46]. Including follow-up post-intervention assessments to assess whether the child continues to make progress with their parents' retained skills would enhance the study's validity by providing valuable additional information.

## Author Contributions

**Conceptualization:** JieYi Png, Masne Kadar.

**Data curation:** JieYi Png.

**Formal analysis:** JieYi Png.

**Funding acquisition:** Farahiyah Wan Yunus.

**Investigation:** JieYi Png.

**Methodology:** JieYi Png.

**Project administration:** JieYi Png.

**Resources:** JieYi Png, Masne Kadar.

**Supervision:** Farahiyah Wan Yunus, Masne Kadar, Yang Wai Wai, Yazmin Ahmad Rusli, Jamilah Hanum Abdul Khaiyom.

**Validation:** Farahiyah Wan Yunus, Masne Kadar.

**Writing – original draft:** JieYi Png.

**Writing – review & editing:** JieYi Png, Farahiyah Wan Yunus, Masne Kadar.

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
