## [Decision Letter · Decision Letter 0]

11 Mar 2024

PONE-D-23-43595Parents’ Online Coaching in the Early Intervention Home Program for Toddlers with Autism Spectrum Disorder During the COVID-19 Pandemic: Manual Development and Feasibility StudyPLOS ONE

Dear Dr. Wan Yunus,

Thank you for submitting your manuscript to PLOS ONE. After careful consideration, we feel that it has merit but does not fully meet PLOS ONE’s publication criteria as it currently stands. Therefore, we invite you to submit a revised version of the manuscript that addresses the points raised during the review process.

We look forward to receiving your revised manuscript.

Kind regards,

Amin Nakhostin-Ansari

Academic Editor

PLOS ONE

Journal Requirements:

"This study was funded by Universiti Kebangsaan Malaysia research grant GGPM-2020-023 and TRANSLATIONAL-2019-001/2."

Reviewers' comments:

Reviewer's Responses to Questions

**Comments to the Author**

1. Is the manuscript technically sound, and do the data support the conclusions?

Reviewer #1: Partly

Reviewer #2: Yes

2. Has the statistical analysis been performed appropriately and rigorously? 

Reviewer #1: Yes

Reviewer #2: No

3. Have the authors made all data underlying the findings in their manuscript fully available?

Reviewer #1: No

Reviewer #2: Yes

4. Is the manuscript presented in an intelligible fashion and written in standard English?

Reviewer #1: Yes

Reviewer #2: Yes

5. Review Comments to the Author

Reviewer #1: Here are some of my comments, one at the end is the major one:

one in 150 children, and 1 in 54 - Are there across the world or US. Please mention it explicitly where this prevalence rate are from. And cite the right references.

Page 4 - line 71: corelated - correlated

getting a diagnosis is reported as 21.5 - What is 21.5 here? Is that age, months, years?

I could read the rest of the paper without any difficulty and the manual development followed good necessary steps. But one of my main concern is the validity of this manual. The feasibility study was conducted only 8 participants putting this work at borderline. If this work has to reach out to people to really change the lives of many toddler with ASD, the current validation with 8 parents/participants is not enough. I may suggest the authors to go for a short paper rather than a full paper. If this work is about to be published, then I would suggest the authors to mention in the number of participant in the abstract and also clearly state that this is just a preliminary study and this manual will not be supplied until they conduct a large study with handful of participants.

Reviewer #2: This manuscript presents a well-designed study aimed at developing and testing an early home based intervention program for toddlers with ASD, with strengths including clear objectives, and positive parental feedback. However, there are some weaknesses that needs to be addressed as mentioned below:

1. There is a need for a control group with no parent coaching that will validate the benefits of the intervention program developed in this study.

2. Are there any other home-based intervention programs that can be compared with the results of this study?

3. Most of the results are qualitative with no statistical analysis. Incorporating quantitative data with statistical analysis will strengthen the results of this study.

4. On page 14, line 276 please expand and clarify “this indicated that the program manual showed excellent content validity”. Most of the readers are not familiar with this value.

5. The limitations of data collection through online platforms especially involving ASD population should be thoroughly discussed.

6. There is a need to update the references with the relevant citations from the year 2023 and 2024.

6. PLOS authors have the option to publish the peer review history of their article (what does this mean?). If published, this will include your full peer review and any attached files.

Reviewer #1: **Yes: **Pradeep Raj Krishnappa Babu

Reviewer #2: No

---

## [Author Response · Author response to Decision Letter 0]

29 Apr 2024

Please see the attached files for respond to reviewers revisions.

---

## [Decision Letter · Decision Letter 1]

14 Jun 2024

PONE-D-23-43595R1Parents’ Online Coaching in the Early Intervention Home Program for Toddlers with Autism Spectrum Disorder During the COVID-19 Pandemic: Manual Development and Feasibility StudyPLOS ONE

Dear Dr. Wan Yunus,

Thank you for submitting your manuscript to PLOS ONE. After careful consideration, we feel that it has merit but does not fully meet PLOS ONE’s publication criteria as it currently stands. Therefore, we invite you to submit a revised version of the manuscript that addresses the points raised during the review process.

**Thank you for revising the manuscript according to the reviewers' comments. While the quality of the manuscript has improved, some reviewers' comments remain unaddressed. Fully addressing all comments is necessary for us to proceed with your paper. Please carefully address all the concerns, especially those raised by reviewer 3, and I would be happy to reconsider your work.**

We look forward to receiving your revised manuscript.

Kind regards,

Amin Nakhostin-Ansari

Academic Editor

PLOS ONE

Reviewers' comments:

Reviewer's Responses to Questions

**Comments to the Author**

1. If the authors have adequately addressed your comments raised in a previous round of review and you feel that this manuscript is now acceptable for publication, you may indicate that here to bypass the “Comments to the Author” section, enter your conflict of interest statement in the “Confidential to Editor” section, and submit your "Accept" recommendation.

Reviewer #1: (No Response)

Reviewer #3: (No Response)

2. Is the manuscript technically sound, and do the data support the conclusions?

Reviewer #1: (No Response)

Reviewer #3: Partly

3. Has the statistical analysis been performed appropriately and rigorously? 

Reviewer #1: (No Response)

Reviewer #3: Yes

4. Have the authors made all data underlying the findings in their manuscript fully available?

Reviewer #1: (No Response)

Reviewer #3: Yes

5. Is the manuscript presented in an intelligible fashion and written in standard English?

Reviewer #1: (No Response)

Reviewer #3: Yes

6. Review Comments to the Author

**Reviewer #1: **Going back to my previous concern, comment 4 - The authors has to clearly mention in the abstract and the objective that this study is just a feasibility study done with 8 parents/participants, which is still missing. This would help a reader to be aware of it right away when they think about the findings of this study, or other researchers would get a hint that they want to validate this scale on a different population before using it.

**Reviewer #3:** Dear Authors,

I appreciate the opportunity to review your paper titled "Parents’ Online Coaching in the Early Intervention Home Program for Toddlers with Autism Spectrum Disorder During the COVID-19 Pandemic: Manual Development and Feasibility Study". Your work addresses an important and timely topic, and you have made significant efforts in developing the manual.

However, I have several major concerns that lead me to recommend rejection of this manuscript in its current form. Firstly, the evaluation of the feasibility of the manual was conducted with a sample size of only 8 parents, which is insufficient to draw meaningful conclusions. Secondly, the absence of a control group limits the ability to evaluate the effectiveness of your manual in comparison to conventional parenting methods. Lastly, the qualitative nature of your study makes it challenging to derive robust and generalizable findings.

I suggest conducting a second phase of research with a larger sample size and incorporating more quantitative measures. This would enhance the rigor of your study and provide stronger evidence for the feasibility and effectiveness of your manual for ASD caregivers.

Thank you for your efforts and consideration.

Sincerely,

7. PLOS authors have the option to publish the peer review history of their article (what does this mean?). If published, this will include your full peer review and any attached files.

Reviewer #1: **Yes: **Pradeep Raj

Reviewer #3: **Yes: **Fateme TaghaviZanjani

---

## [Author Response · Author response to Decision Letter 1]

28 Jul 2024

Please see attached the respond to reviewer's word attached.

---

## [Decision Letter · Decision Letter 2]

13 Aug 2024

Parents’ Online Coaching in the Early Intervention Home Program for Toddlers with Autism Spectrum Disorder During the COVID-19 Pandemic: Manual Development and Feasibility Study

PONE-D-23-43595R2

Dear Dr. Yunus,

We’re pleased to inform you that your manuscript has been judged scientifically suitable for publication and will be formally accepted for publication once it meets all outstanding technical requirements.

Kind regards,

Amin Nakhostin-Ansari

Academic Editor

PLOS ONE

Additional Editor Comments (optional):

Reviewers' comments:

Reviewer's Responses to Questions

**Comments to the Author**

1. If the authors have adequately addressed your comments raised in a previous round of review and you feel that this manuscript is now acceptable for publication, you may indicate that here to bypass the “Comments to the Author” section, enter your conflict of interest statement in the “Confidential to Editor” section, and submit your "Accept" recommendation.

Reviewer #1: All comments have been addressed

Reviewer #3: All comments have been addressed

2. Is the manuscript technically sound, and do the data support the conclusions?

Reviewer #1: Yes

Reviewer #3: Yes

3. Has the statistical analysis been performed appropriately and rigorously? 

Reviewer #1: Yes

Reviewer #3: I Don't Know

4. Have the authors made all data underlying the findings in their manuscript fully available?

Reviewer #1: No

Reviewer #3: Yes

5. Is the manuscript presented in an intelligible fashion and written in standard English?

Reviewer #1: Yes

Reviewer #3: Yes

6. Review Comments to the Author

Reviewer #1: (No Response)

Reviewer #3: Dear authors,

I hope this message finds you well. Firstly, I appreciate your thorough responses to my initial comments and concerns. Your explanations regarding the nature of a feasibility study and the inherent limitations such as a small sample size and the absence of a control group were well-articulated and satisfactory. The revisions made to the manuscript significantly improve its clarity and rigor, with a compelling emphasis on the study's feasibility objectives and the potential for future research in this area.

I am pleased to inform you that I have recommended the acceptance of your manuscript to the editorial board as your work provides a crucial foundation for future studies.

Thank you for addressing my feedback so effectively. I look forward to seeing your study published and wish you continued success in your research endeavors.

Best regards,

7. PLOS authors have the option to publish the peer review history of their article (what does this mean?). If published, this will include your full peer review and any attached files.

Reviewer #1: **Yes: **Pradeep Raj K B

Reviewer #3: **Yes: **Fateme TaghaviZanjani

---

## [Editor Report · Acceptance letter]

16 Aug 2024

PONE-D-23-43595R2 

PLOS ONE

Dear Dr. Wan Yunus, 

I'm pleased to inform you that your manuscript has been deemed suitable for publication in PLOS ONE. Congratulations! Your manuscript is now being handed over to our production team.

Kind regards, 

on behalf of

Dr. Amin Nakhostin-Ansari 

Academic Editor

PLOS ONE